# Optimising Network Architectures for Provable Adversarial Robustness

## Abstract

Existing Lipschitz-based provable defences to adversarial examples only cover the $\ell_2$ threat model. We introduce the first bound that makes use of Lipschitz continuity to provide a more general guarantee for threat models based on any $\ell_p$ norm. Additionally, a new strategy is proposed for designing network architectures that exhibit superior provable adversarial robustness over conventional convolutional neural networks. Experiments are conducted to validate our theoretical contributions, show that the assumptions made during the design of our novel architecture hold in practice, and quantify the empirical robustness of several Lipschitz-based adversarial defence methods.

## 1 Introduction

The robustness of deep neural networks to adversarial attack (Szegedy et al., 2014) is an increasingly topical issue as deep models are becoming more widely deployed in practice. This paper focuses on the problem of ensuring that once a deep network trained for image classification has been deployed, one can be confident that an adversary has only a limited ability to maliciously impact model predictions when they tamper with the system inputs. Such malicious inputs, so-called adversarial examples, appear to humans as normal images, but in reality they have undergone imperceptible modifications that cause a model to make an incorrect prediction. In contrast to many empirically focused approaches to defending against adversarial interference (see Carlini & Wagner (2017) and Athalye et al. (2018) for more details), we look to the trends prevalent in cryptography research, where developing provable defences is the primary modus operandi. The analysis in this paper makes use of the Lipschitz properties of neural networks to accomplish this.

We begin by extending existing theory addressing the relationship between Lipschitz continuity and provable adversarial robustness. Using insights from the resulting bounds, it is shown how one can adjust network architectures in such a way that Lipschitz-based regularisation methods are more effective. Experimental results show that, while having little difference in empirical adversarial robustness compared to existing Lipschitz-based defences, our approach improves the level of *provable* robustness significantly. We also provide evidence that indicates adversarial training has only a small penalising effect on the upper bound of the Lipschitz constant of a network currently used by most works.

## 2 Related Work

There are several papers in the literature on deep learning that address adversarial robustness through the use of Lipschitz continuity, however they focus solely on perturbations with bounded Euclidean norm. Tsuzuku et al. (2018) present an efficient method for determining whether an example could have been tampered with at test time or, conversely, certify that a prediction has not been influenced by an adversarial attack. They compare the prediction margin normalised by the Lipschitz constant of the network to the magnitude of the largest perturbation allowed by the threat model, allowing them to determine whether the input could be an adversarial example. Farnia et al. (2018) show how the adversarial risk can be bounded in terms of the training loss by adapting the bound of Bartlett et al. (2017) to consider perturbations to the margin, using a similar technique to Tsuzuku et al. (2018).

The analysis in this paper takes a similar high-level strategy—making use of margins and Lipschitz constants—but we extend this theory to threat models based on arbitrary $p$-norms, and provide a simpler proof than Tsuzuku et al. (2018). Huster et al. (2018) demonstrate that current methods for regularising Lipschitz constants of networks have deficiencies when used for improving adversarial robustness. Specifically, it is shown that existing approaches for regularising the Lipschitz constant may be too restrictive because the bound on the Lipschitz constant is too loose, resulting in over-regularisation. In anticipation that future work can provide improved upper bounds on the Lipschitz constant of a network, we provide theoretical and practical contributions that are compatible with arbitrary bounds.

Existing work that aims to provide theory-backed guarantees for adversarial robustness has resulted in several techniques able to certify whether a prediction for a particular example is immune to adversarial attack under a threat model based on $\ell_p$-norm perturbation size. Weng et al. (2018) propose a method that can only be applied to networks composed of fully connected layers with rectified linear units activation functions, and no batch normalisation. Wong & Kolter (2018) present an approach based on solving an optimisation problem. While the robustness estimates they give are considerably tighter than many other certification methods, they scale very poorly to networks with large input images or feature maps. In contrast to these methods, our approach bounds the expected adversarial generalisation error, has virtually no test-time computational overhead, and can be applied to arbitrary feed-forward architectures. Bounding the expected generalisation error enables us to give guarantees about the level of robustness a model will have once it has been deployed. Existing approaches to provable robustness do not come with such guarantees, and can only provide certification for individual instances.

## 3  GENERALISATION UNDER ATTACK

Methods for estimating the generalisation performance of learned models typically assume examples, $(\vec{x}, y)$, observed at both training and testing time are independently drawn from the same distribution, $\mathcal{D}$. Such methods estimate or bound the expected risk,

$$R^\ell(f) = \mathbb{E}_{(\vec{x}, y) \sim \mathcal{D}}[\ell(f(\vec{x}), y)], \tag{1}$$

of a classifier, $f$, with respect to some loss function, $\ell$. The standard technique for estimating the expected risk in deep learning is to use an empirical approximation measured on a set of held-out data. Literature on statistical learning theory often investigates the related problem of quantifying worst-case generalisation performance. This is typically accomplished by deriving bounds expressed in terms of the empirical risk measured on the training data, and a term related to some measure of complexity of the hypothesis class.

In the adversarial setting one must consider the expected risk when under the influence of an attacker that can add perturbations to feature vectors at test time,

$$\tilde{R}^\ell_{p,t}(f) = \mathbb{E}_{(\vec{x}, y) \sim \mathcal{D}} \left[ \max_{\vec{\epsilon} : \|\vec{\epsilon}\|_p \leq t} \ell(f(\vec{x} + \vec{\epsilon}), y) \right], \tag{2}$$

which is known as the adversarial risk (Madry et al., 2018). In contrast to the expected risk, $\tilde{R}^{\mathcal{L}}_{p,t}(f)$ cannot be reliably approximated from data when $f$ is non-convex, as one must find the globally optimal setting of $\vec{\epsilon}$ for each data point in the held-out set.

For a hypothesis, $f$, that produces a vector of real-valued scores, each associated with a possible class, we define the margin function as

$$m_f(\vec{x}, y) = f_y(\vec{x}) - \max_{j \neq y} f_j(\vec{x}), \tag{3}$$

where $f_i(\vec{x})$ is the $i$th component of the output of $f(\vec{x})$. Typical loss functions for measuring the performance a model via composition with the margin function include the zero–one loss and the hinge. These compositions result in the classification error rate and the multi-class hinge loss variant proposed by Crammer & Singer (2001), respectively.

**Proposition 1.** *If $f$ is $k$-Lipschitz with respect to the p-norm and $\ell : \mathbb{R} \to \mathbb{R}^+$ is a monotonically decreasing loss function, then*

$$\max_{\vec{\epsilon} : \|\vec{\epsilon}\|_p \leq t} \ell(f(\vec{x} + \vec{\epsilon}), y) \leq \ell(m_f(\vec{x}, y) - 2^{1/q} kt), \tag{4}$$

*where $q$ is defined such that $\|\cdot\|_q$ is the dual norm of $\|\cdot\|_p$.*

*Proof.* Note that one can define the margin function given in Equation 3 as $m_f(\vec{x}, y) = m_{\mathbb{I}}(f(\vec{x}), y)$, where $\mathbb{I}$ is the identity function. The Lipschitz constant of $m_{\mathbb{I}}$ with respect to its first argument when using the $p$-norm is $\max_{\vec{x}} \|\nabla_{\vec{x}} m_{\mathbb{I}}(\vec{x}, y)\|_q$ (Shalev-Shwartz, 2012, p. 133). The gradient of $m_{\mathbb{I}}$ is a vector with all elements set to zero, except for those corresponding to the largest and second largest components of $\vec{x}$. These components of the gradient take the values of 1 and $-1$, respectively. Plugging these values into the definition of vector $p$-norms, one arrives at a Lipschitz constant of $2^{1/q}$. From the composition property of Lipschitz functions, we can say that $m_f$ is $(2^{1/q}k)$-Lipschitz with respect to $\vec{x}$. The Lipschitz property of $m_f$ can be used to bound the worst-case change in the output the margin function for a bounded change in the input, yielding

$$\ell(\min_{\vec{\epsilon}:\|\vec{\epsilon}\|<t} m_f(\vec{x} + \vec{\epsilon}, y)) \leq \ell(m_f(\vec{x}, y) - 2^{1/q}kt). \tag{5}$$

From the decreasing monotonicity of $\ell$, we have that

$$\max_{\vec{\epsilon}:\|\vec{\epsilon}\|<t} \ell(m_f(\vec{x} + \vec{\epsilon}, y)) = \ell(\min_{\vec{\epsilon}:\|\vec{\epsilon}\|<t} m_f(\vec{x} + \vec{\epsilon}, y)), \tag{6}$$

which concludes the proof. $\qquad\square$

This proposition bounds the worst-case change in loss for a single image in terms of prediction confidence, Lipschitz constant of the network, and the maximum allowable attack strength.

The relationship given in Proposition 1 is a more general form of the bound derived by Tsuzuku et al. (2018), who consider only the Euclidean norm. Gouk et al. (2018) provide efficient techniques for computing upper bounds to the Lipschitz constant of feed-forward networks with respect to the 1-, 2-, and $\infty$-norms. Golowich et al. (2018) present a bound on the Rademacher complexity of deep networks that is primarily dependent on the same quantity that Gouk et al. (2018) show to be an upper bound on the $\infty$-norm Lipschitz constant. They do not make use of Lipschitz properties to prove this, so tighter bounds on the Lipschitz constant will not necessarily improve the bound on Rademacher complexity. These two pieces of related work, coupled with Proposition 1, imply that empirical risk minimisation with a Lipschitz-based regularisation method is a principled approach to controlling adversarial risk.

Proposition 1 can be extended to provide a non-trivial bound on the expected adversarial risk through the use of a held-out dataset and a simple application of McDiarmid's inequality. Although this is not a standard approach to take in learning theoretic generalisation bounds, it is a practical tool that can be used for quantifying worst-case adversarial robustness that is of great utility to practitioners.

**Proposition 2.** *If $f$ is $k$-Lipschitz with respect to the $p$-norm, $\ell : \mathbb{R} \to [0, B)$ is a monotonically decreasing loss function, and $\{(\vec{x}_i, y_i) \sim \mathcal{D}\}_{i=1}^n$ is independent of $f$ (i.e., held-out data), the following holds with probability at least $1 - \delta$:*

$$\tilde{R}_{p,t}^{\ell}(f) \leq \frac{1}{n} \sum_{i=1}^n \ell(m_f(\vec{x}_i, y_i) - 2^{1/q}kt) + B\sqrt{\frac{\ln(2/\delta)}{2n}} \tag{7}$$

*where $q$ is defined such that $\|\cdot\|_q$ is the dual norm of $\|\cdot\|_p$.*

*Proof.* Constructing a mean over loss terms applied to i.i.d. samples from $\mathcal{D}$,

$$L = \frac{1}{n} \sum_{i=1}^n \ell(m_f(\vec{x}_i, y_i) - 2^{1/q}kt), \tag{8}$$

results in a Doob martingale where each term is bounded by $\frac{1}{B}$. This allows McDiarmid's inequality to bound the probability of the difference between the summation and its expected value exceeding some threshold,

$$\mathbb{P}(|L - \mathbb{E}[L]| > \gamma) \leq 2\exp\left(\frac{-2n\gamma^2}{B^2}\right). \tag{9}$$

Setting $\delta$ equal to the right-hand side of Inequality 9 and solving for $\gamma$ yields

$$\gamma = B\sqrt{\frac{\ln(2/\delta)}{2n}}. \tag{10}$$

Thus, we can say with confidence $1 - \delta$ that

$$\mathbb{E}[L] \leq L + B\sqrt{\frac{\ln(2/\delta)}{2n}}. \tag{11}$$

Applying Proposition 1 to each term of the summation, $L$, concludes the proof. □

Proposition 2 extends the result of Proposition 1 from the loss on a single instance to the expected risk.

In practice, this means that a practitioner can bound the worst-case adversarial performance of their model based on its (non-adversarial) validation-set performance and its Lipschitz constant, both of which can be measured efficiently. As we show later, this can lead to non-vacuous bounds on error rate, which in turn could allow a user to deploy a model with provable confidence about its performance under adversarial attack—without the hassle and computational expense of instance-wise certification at run-time (Weng et al., 2018; Wong & Kolter, 2018).

## 4 Architectures for Provable Robustness

The analysis in Section 3 motivates a high-level strategy for improving the adversarial robustness of neural networks: maximise the prediction margin while minimising the Lipschitz constant of the model. Several papers have proposed different methods for regularising the Lipschitz constant of a network, with various motivations, including improving robustness to adversarial exmaples (Tsuzuku et al., 2018; Cisse et al., 2017) and improving generalisation performance in the non-adversarial case (Gouk et al., 2018).

We propose a strategy for modifying network architectures to make them more amenable to Lipschitz-based regularisers: splitting a single multi-class classification network into a collection of one-versus-all (OVA) classifiers that each produce a real-valued score. Unlike the conventional OVA method, where each component classifier is trained in isolation, the networks used in our approach are still trained jointly using a softmax composed with the cross entropy loss function. The core assumption this approach is predicated on is that each of the binary classifiers needs to solve a simpler problem than the original multi-class classification task. Gouk et al. (2018) show that the Lipschitz constant is related to model capacity, so the subnetwork associated with each class should be able to achieve high accuracy with a smaller Lipschitz constant than a conventional multi-class classification network. This makes such an architecture more amenable to Lipschitz-based regularisation, and hence provable robustness via Proposition 2.

Consider the vector-valued function,

$$f(\vec{x}) = [f_1(\vec{x}), f_2(\vec{x}), ..., f_C(\vec{x})], \tag{12}$$

where $C$ is the number of classes, and $f_i$ is $k_i$-Lipschitz. We have from the Lipschitz property of each $f_i$ that

$$\|f(\vec{x}) - f(\vec{x} + \vec{v})\|_p \leq \|[k_1\|\vec{v}\|_p, k_2\|\vec{v}\|_p, ..., k_C\|\vec{v}\|_p]\|_p \tag{13}$$

$$= \|\vec{v}\|_p\|[k_1, k_2, ..., k_C]\|_p, \tag{14}$$

from which we can deduce that the Lipschitz constant of the one-versus-all classifier is the $\ell_p$ norm of the vector of Lipschitz constants corresponding to each binary classifier. In the case of the $\infty$-norm, the largest Lipschitz constant associated with a single binary classifier dictates the Lipschitz constant of the entire OVA classifier.

### 4.1 Lipschitz Regularization Training

We investigate two approaches to controlling the Lipschitz constant of neural networks. The first approach we use is to add the bound on the Lipschitz constant as a regularisation term to the objective function, resulting in

$$\frac{1}{n} \sum_{i=1}^{n} \ell(f(\vec{x}_i), y_i) + \prod_{l=1}^{d} \|W_l\|_p, \tag{15}$$

where $d$ is the number of layers in the network and $\|\cdot\|_p$ is the operator norm induced by the vector $p$-norm. In the case where $p$ is two, the operator norm is the largest singular value of the matrix (i.e., the spectral norm). For $p = \infty$, it is the maximum absolute row sum norm (Gouk et al., 2018),

$$\|W\|_\infty = \max_i \sum_j |W_{i,j}|. \tag{16}$$

The other regularisation technique we consider is the method proposed by Tsuzuku et al. (2018), which can be adapted to regularise the $\ell_\infty$ Lipschitz constant by using the norm in Equation 16, rather than the spectral norm.

## 5 Experiments

This section presents the results of numerical experiments that demonstrate the tightness of the bounds presented in Section 3, provides evidence that the architecture proposed in Section 4 is inherently easier to optimise for provable robustness than conventional network architectures, and evaluates the empirical robustness of Lipschitz-based defence methods. The models used in these experiments were implemented using Keras (Chollet, 2015), and the adversarial attacks were performed using the CleverHans toolkit (Papernot et al., 2016).

### 5.1 Tightness of the Bound

The bound given in Proposition 2 provides a way to estimate the worst-case performance of a model when under the influence of an adversary. In order to validate this bound empirically, we train linear support vector machines with different levels of $\ell_2$ regularisation on the MNIST dataset of hand-written digits. In the case of linear SVMs, the optimisation problem solved by iterative gradient-based attacks, such as the projected gradient descent method of Madry et al. (2018), are convex and can therefore be solved globally. This means the the empirical adversarial risk can be computed exactly. Plots indicating the tightness of the bound for linear SVMs are given in Figures 1 and 2. These were generated by training models on the first 50,000 instances of the training set, using the other 10,000 training instances as the held-out data required for computing the bound, and using the PGD attack when evaluating the network on the test data. These plots confirm that the bound proposed in Proposition 2 is non-vacuous and has the potential to be useful in practice.

### 5.2 Provable Robustness

We first experiment on MNIST to determining whether our proposed OVA networks achieve better provable robustness than conventional convolutional neural networks. To control for the potentially confounding factor of model capacity, a series of networks with different widths are trained. We define the width of a conventional convolutional network as the number of feature maps produced by the first convolutional layer. For OVA networks, the width is the number of feature maps produced by the first layer in a single binary classifier, multiplied by the number of binary classifiers. For both network types, the chosen architectures contain two convolutional layers, the second of which has twice the number of feature maps as the first. Each convolutional layer contains $5 \times 5$ kernels, rectified linear unit activation functions, and is followed by a $2 \times 2$ max pooling layer. After the convolutional layers are two fully connected layers: one with 128 hidden units, and another with either 10 units (for conventional networks), or one unit (for OVA networks).

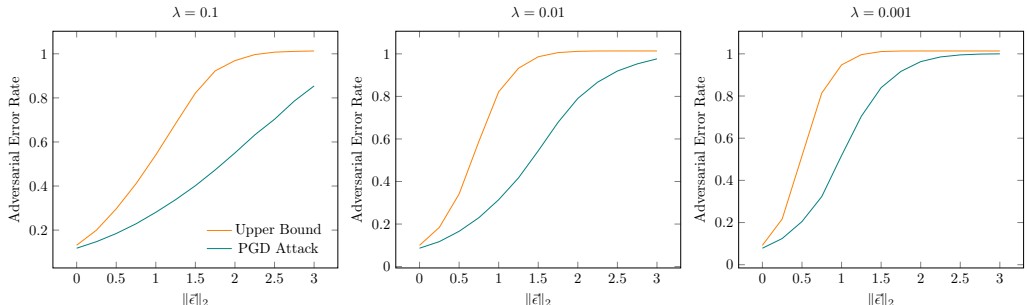

Figure 1: Plots demonstrating the relationship between the provable upper bound on adversarial risk, and the actual misclassification rate on the test set under adversarial attack. Linear SVM recognition on MNIST with $\ell_2$ threat model and regularization strength $\lambda$.

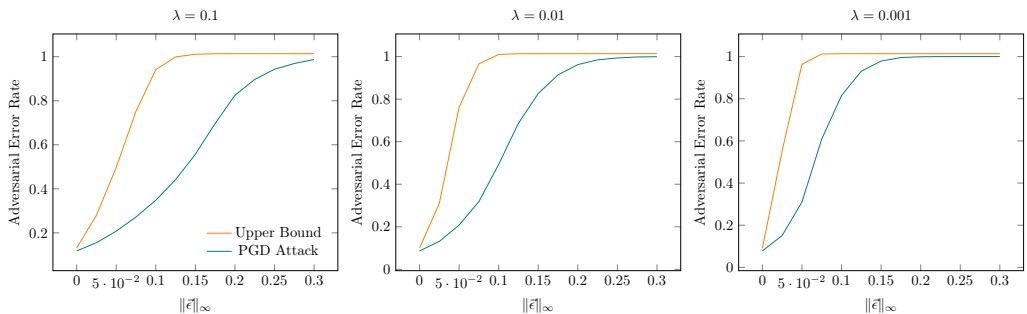

Figure 2: Plots demonstrating the relationship between the provable upper bound on adversarial risk, and the actual misclassification rate on the test set under adversarial attack. Linear SVM recognition on MNIST with $\ell_\infty$ threat model and regularization strength $\lambda$.

Figures 3 and 4 show how the number of feature maps in each layer of the networks impacts the provable adversarial robustness for the $\ell_2$ and $\ell_\infty$ norms under different threat models, respectively. The models in these plots are regularised using the Lipschitz penalty method proposed in Section 4. These figures show that: (1) Regularised OVA networks exhibit superior provable robustness compared to regularized conventional CNNs at comparable model sizes, (2) The magnitude of this margin becomes more pronounced as model size increases, (3) All methods have low error rate for unperturbed examples (left plots).

To investigate how well OVANets scale, additional experiments are run on the CIFAR-10 dataset, using VGG-style networks (Simonyan & Zisserman, 2014) as the base architecture. The baseline CNN uses the VGG11 architecture, and each subnetwork of the OVANet architecture is a VGG11 network with half the number of feature maps in each layer. Table 1 provides probabilistic (95% confidence) bounds on the worst-case adversarial error rate using Proposition 2. Table 2 shows the corresponding provable robustness results for SVHN benchmark. From the results we can see that: (1) Lipschitz penalty training improves the adversarial error rate for both vanilla VGG11 and VGG11-OVANet (performance improves with $\lambda$); (2) VGG11-OVANet generally has superior provable robustness compared to vanilla VGG11 for corresponding regularisation strength, especially for strong attacks. (3) Meanwhile, regularized OVANet achieves comparable results to a regularized CNN in terms of clean data performance.

### 5.3 EMPIRICAL ROBUSTNESS

OVA networks exhibit superior provable robustness compared to conventional networks, but it does not immediately follow from this that their empirical robustness will also be improved. We investigate this by using the PGD attack of Madry et al. (2018) to attack networks defended with Lipschitz Margin Training (LMT), the Lipschitz penalty regulariser (LP), and

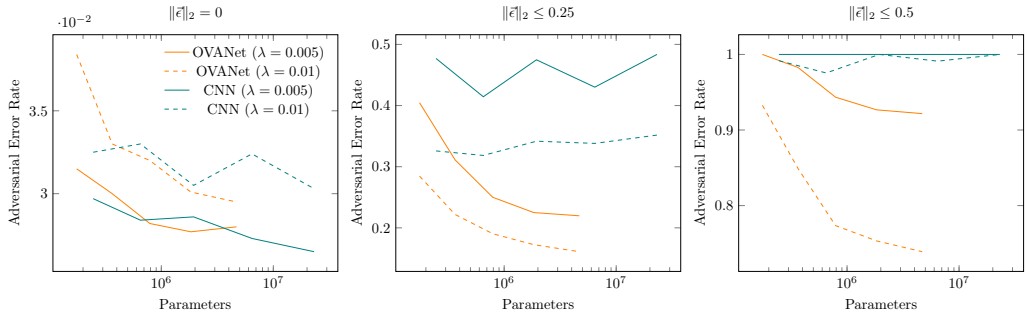

Figure 3: Comparison of provable adversarial risk for conventional CNN versus OVANet trained with Lipschitz penalty regularization over a range of model sizes. The $\ell_2$ Lipschitz constant is used for regularisation and computing the bound. OVANet shows superior provable robustness, especially at larger model sizes.

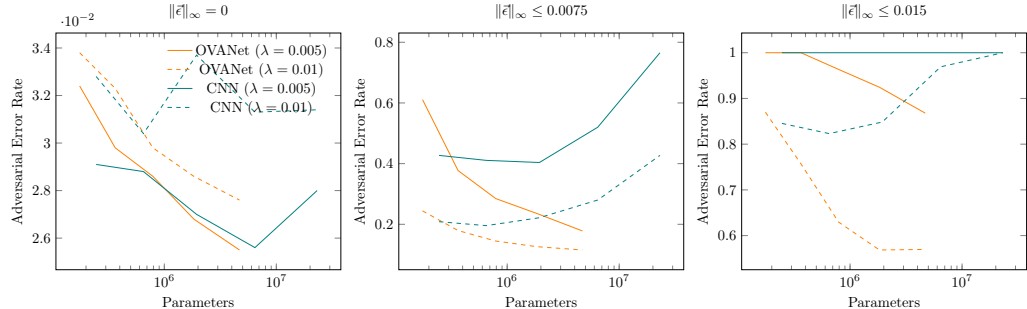

Figure 4: Comparison of provable adversarial risk for conventional CNN versus OVANet trained with Lipschitz penalty regularization over a range of model sizes. The $\ell_\infty$ Lipschitz constant is used for regularisation and computing the bound. OVANet shows superior provable robustness, especially at larger model sizes.

no defence. For each method, the best performing architecture is chosen by measuring the empirical robustness of the validation set. The set of architectures considered are the same as those discussed Figures 3 and 4. The results are given in Tables 3 and 4 for the $\ell_2$ and $\ell_\infty$ norms, respectively.

For the $\ell_2$ norm, the OVA networks perform similarly to their conventional counterparts from an empirical standpoint, but with improved provable robustness. For the $\ell_\infty$ norm, we do not observe an empirical benefit from using OVA networks over conventional network architectures. However, our adaptation of Lipschitz-based regularisation to the $\ell_\infty$ setting does improve empirical robustness noticeably over the undefended network—particularly when using our novel penalty approach given in Equation 15, rather than LMT.

Table 5 shows the upper bound on the Lipschitz constants of each model for which we have reported empirical robustness measurements. From this table, we can see that both Lipschitz-based defences successfully limit the constant of each network—sometimes by several orders of magnitude. Using an OVA network architecture, in most cases, results in another modest decrease in the Lipschitz constant when using a Lipschitz regularisation method, and a substantial decrease when no regulariser is applied.

## 6   Conclusions

This paper presents a $p$-norm-agnostic theoretical analysis of provable adversarial robustness via Lipschitz regularisation. A new architecture, the OVA network, is proposed, motivated by insights of how Lipschitz constants can be bounded for different architecture design choices. It is shown that OVA networks achieve similar empirical robustness to conventional neural

| Model | $\lambda$ | Clean | Perturbation Size ($\ell_2$) | | | |
|---|---|---|---|---|---|---|
| | | | 1/255 | 2/255 | 3/255 | 4/255 |
| VGG11-CNN | 0 | 14.50 | 100.00 | 100.00 | 100.00 | 100.00 |
| | 0.0001 | 14.22 | 47.61 | 79.22 | 95.87 | 100.00 |
| | 0.0005 | 16.00 | 29.00 | 42.74 | 56.49 | 69.84 |
| | 0.001 | 17.64 | 26.80 | 35.60 | 44.75 | 53.66 |
| VGG11-OVANet | 0 | 17.18 | 100.00 | 100.00 | 100.00 | 100.00 |
| | 0.0001 | 15.58 | 44.54 | 73.49 | 93.11 | 99.99 |
| | 0.0005 | 15.86 | 27.68 | 39.01 | 51.85 | 63.68 |
| | 0.001 | 17.09 | 25.00 | 32.54 | 40.35 | 48.53 |

Table 1: Bounds on the error rate for VGG models trained on CIFAR-10 with the Lipschitz penalty regulariser. The bounds were computed with Proposition 2 at the 95% confidence level and the $\ell_2$ threat model.

| Model | $\lambda$ | Clean | Perturbation Size ($\ell_2$) | | | |
|---|---|---|---|---|---|---|
| | | | 1/255 | 2/255 | 3/255 | 4/255 |
| VGG11-CNN | 0 | 7.29 | 100.00 | 100.00 | 100.00 | 100.00 |
| | 0.0001 | 7.15 | 13.16 | 21.86 | 33.59 | 47.10 |
| | 0.0005 | 8.46 | 11.38 | 14.84 | 19.33 | 24.49 |
| | 0.001 | 9.41 | 11.76 | 14.95 | 17.76 | 21.48 |
| VGG11-OVANet | 0 | 7.69 | 100.00 | 100.00 | 100.00 | 100.00 |
| | 0.0001 | 7.45 | 12.10 | 19.19 | 28.38 | 39.17 |
| | 0.0005 | 8.25 | 10.64 | 13.02 | 16.11 | 19.82 |
| | 0.001 | 9.02 | 10.92 | 12.86 | 15.05 | 17.81 |

Table 2: Bounds on the error rate for VGG models trained on SVHN with the Lipschitz penalty regulariser. The bounds were computed with Proposition 2 at the 95% confidence level and the $\ell_2$ threat model.

networks when Lipschitz-based defences are used. However, as network size increases, OVA networks are able to achieve significantly better certifiable robustness. This is a useful result for practitioners, who can use our LP regulariser and bound in order to train models with quantitatively certifiable degree of robustness against adversarial attack.

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

| | | Perturbation Size ($\ell_2$) | | | |
|---|---|---|---|---|---|
| Model | Clean | 0.5 | 1.0 | 1.5 | 2.0 |
| CNN | 0.80 | 2.87 | 14.90 | 49.64 | 74.73 |
| CNN + LMT | 1.37 | 2.97 | 10.24 | 29.94 | 65.22 |
| CNN + LP | 1.72 | 3.51 | 10.94 | 29.66 | 61.80 |
| OVANet | 0.90 | 3.63 | 17.91 | 55.88 | 82.82 |
| OVANet + LMT | 1.34 | 3.23 | 11.18 | 32.55 | 65.49 |
| OVANet + LP | 1.90 | 3.44 | 11.42 | 30.51 | 58.50 |

Table 3: Error rates (lower is better) of models trained with different regularisers when attacked with the projected gradient descent method of Madry et al. (2018). The $\ell_2$ threat model and regularisers are used. While our OVANet architecture does not improve empirical robustness, our LP training does benefit both model architectures.

| | | Perturbation Size ($\ell_\infty$) | | |
|---|---|---|---|---|
| Model | Clean | 0.1 | 0.2 | 0.3 |
| CNN | 0.80 | 26.72 | 94.66 | 99.34 |
| CNN + LMT | 1.13 | 16.51 | 85.90 | 99.03 |
| CNN + LP | 1.52 | 14.40 | 79.34 | 98.94 |
| OVANet | 0.90 | 36.42 | 97.52 | 99.37 |
| OVANet + LMT | 1.00 | 21.48 | 93.01 | 99.29 |
| OVANet + LP | 1.25 | 17.32 | 88.63 | 99.10 |

Table 4: Empirical error rates (lower is better) of models trained with different regularisers when attacked with the projected gradient descent method of Madry et al. (2018). The $\ell_\infty$ threat model and regularisers are used. While our OAVNet architecture does not improve empirical robustness, our LP training does benefit both model architectures.

| | Lipschitz Constant | |
|---|---|---|
| Model | $\ell_2$ | $\ell_\infty$ |
| CNN | 361.0 | 148,886.8 |
| CNN + LMT | 23.2 | 711.7 |
| CNN + LP | 11.3 | 261.2 |
| OVANet | 190.9 | 21,957.0 |
| OVANet + LMT | 22.7 | 741.3 |
| OVANet + LP | 9.4 | 215.8 |

Table 5: Lipschitz constants for different architecture and regulariser combinations.

Koby Crammer and Yoram Singer. On the Algorithmic Implementation of Multiclass Kernel-based Vector Machines. *Journal of Machine Learning Research*, 2(Dec):265–292, 2001. ISSN ISSN 1533-7928.

Farzan Farnia, Jesse M. Zhang, and David Tse. Generalizable Adversarial Training via Spectral Normalization. *arXiv:1811.07457 [cs, stat]*, November 2018.

Noah Golowich, Alexander Rakhlin, and Ohad Shamir. Size-Independent Sample Complexity of Neural Networks. In *Conference On Learning Theory*, pp. 297–299, July 2018.

Henry Gouk, Eibe Frank, Bernhard Pfahringer, and Michael Cree. Regularisation of Neural Networks by Enforcing Lipschitz Continuity. *University of Waikato Technical Report*, 2018.

Todd Huster, Cho-Yu Jason Chiang, and Ritu Chadha. Limitations of the Lipschitz constant as a defense against adversarial examples. *arXiv:1807.09705 [cs, stat]*, July 2018.

Aleksander Madry, Aleksandar Makelov, Ludwig Schmidt, Dimitris Tsipras, and Adrian Vladu. Towards Deep Learning Models Resistant to Adversarial Attacks. In *International Conference on Learning Representations*, February 2018.

Nicolas Papernot, Fartash Faghri, Nicholas Carlini, Ian Goodfellow, Reuben Feinman, Alexey Kurakin, Cihang Xie, Yash Sharma, Tom Brown, Aurko Roy, Alexander Matyasko, Vahid Behzadan, Karen Hambardzumyan, Zhishuai Zhang, Yi-Lin Juang, Zhi Li, Ryan Sheatsley, Abhibhav Garg, Jonathan Uesato, Willi Gierke, Yinpeng Dong, David Berthelot, Paul Hendricks, Jonas Rauber, Rujun Long, and Patrick McDaniel. Technical Report on the CleverHans v2.1.0 Adversarial Examples Library. *arXiv:1610.00768 [cs, stat]*, October 2016.

Shai Shalev-Shwartz. Online Learning and Online Convex Optimization. *Foundations and Trends® in Machine Learning*, 4(2):107–194, March 2012. ISSN 1935-8237, 1935-8245. doi: 10.1561/2200000018.

Karen Simonyan and Andrew Zisserman. Very deep convolutional networks for large-scale image recognition. *arXiv preprint arXiv:1409.1556*, 2014.

Christian Szegedy, Wojciech Zaremba, Ilya Sutskever, Joan Bruna, Dumitru Erhan, Ian Goodfellow, and Rob Fergus. Intriguing properties of neural networks. In *arXiv:1312.6199 [Cs]*, 2014.

Yusuke Tsuzuku, Issei Sato, and Masashi Sugiyama. Lipschitz-Margin Training: Scalable Certification of Perturbation Invariance for Deep Neural Networks. In *Advances in Neural Information Processing Systems*, 2018.

Tsui-Wei Weng, Huan Zhang, Hongge Chen, Zhao Song, Cho-Jui Hsieh, Duane Boning, Inderjit S Dhillon, and Luca Daniel. Towards fast computation of certified robustness for relu networks. In *International Conference on Machine Learning*, 2018.

Eric Wong and J. Zico Kolter. Provable defenses against adversarial examples via the convex outer adversarial polytope. In *International Conference on Machine Learning*, 2018.

