# OpenReview forum: "Optimising Neural Network Architectures for Provable Adversarial Robustness"
_ICLR.cc/2020/Conference — Reject_

### Official Review · AnonReviewer2 · 2019-10-22
**Official Blind Review #2**

**Rating:** 3

**Review:**

Summary:
The author show that lipschitz constants of the neural network can be used to bound adversarial robustness. They then propose an adjustment to the network architecture that, according to their assumption, would result in smaller lipschitz constant, which according to the theory would correlate with better robustness.

Comments:
"A practitioner can bound the worst-case adversarial performance of their model based on its (non-adversarial) validation-set performance and its Lipschitz constant, both of which can be measured efficiently."
-> The Lipschitz constant can be measured efficiently only at the condition that you don't care about getting a tight one.
The method of Gouk et al. is a very loose overapproximation of the Lipschitz constant of the neural network (multiplication of lipschitz constant of each layer).

- What happens if we want to obtain results for the 1-norm. Do all the arguments hold up, given that the dual norm is the infinity norm?

Note: I did not assess the correctness of Proposition 2.

"Unlike the conventional OVA method, where each component classifier is trained in isolation, the networks used in our approach are still trained jointly using a softmax composed with the cross entropy loss function."
How is each network a one vs all if they are trained jointly? Is there an additional loss term on top of the joint softmax? Is it just that up until that point they are different networks?

"The core assumption this approach is predicated on is that each of the binary classifiers needs to solve a simpler problem than the original multi-class classification task."
-> There is no validation of this assumption anywhere. This would improve the paper if there was some signs that the justification proposed by the authors have some experimental validation.

"the Lipschitz constant of the one-versus-all classifier is the `p norm of the vector of Lipschitz constants corresponding to each binary classifier", end of page 4.
-> A lipschitz constant, not THE lipschitz constant. This construction gives a valid lipschitz constant but it's entirely possible and very likely that there is a much tighter one.

- Experiments in 5.1 are made on linear SVMs, which means that the lipschitz constant can be computed exactly but this doesn't really reflect how useful the bound will be on network with hidden layers, for which the lipschitz constant will have to be an approximation. This is a valid toy-example / sanity check but not a good validation of the method.

- Experiments in 5.2 and 5.3 seem to indicate that the change in architecture proposed has at least some effect on the robustness to adversarial examples.

Notation / typos:
- In Proposition 1, on the left side of the inequality, l takes as argument a vector (f(x + eps)) and a coordinate, but on the right side and on the definition of l is a scalar function. Does that mean that the left side is the y-th coordinate? As in, what's in l is f_y?
- Page 4, "exmaples"

Some maybe interesting litterature that the paper might benefit from referencing/discussing:
https://openreview.net/forum?id=HJguLo0cKQ, submitted to last years ICLR observed that ensemble of small models tended to be more robust than large models. I wonder if the reasoning in this paper might explain it at least partially.


Opinion:
At the moment, the paper is quite confusing and hard to read, and it's not entirely clear what the crux of the paper, the new architecture is doing. It's also not validated where does the effect comes from.  I'm open to updating my opinion of the paper if these comments are addressed.

**Experience Assessment:**

I have read many papers in this area.

**Review Assessment: Checking Correctness Of Derivations And Theory:**

I assessed the sensibility of the derivations and theory.

**Review Assessment: Checking Correctness Of Experiments:**

I assessed the sensibility of the experiments.

**Review Assessment: Thoroughness In Paper Reading:**

I read the paper at least twice and used my best judgement in assessing the paper.

---

### Official Review · AnonReviewer3 · 2019-10-23
**Official Blind Review #3**

**Rating:** 1

**Review:**

Summary:

This paper presents interesting theoretical contributions towards measuring the robustness of Lipschitz constrained neural networks. The empirical results are fairly limited and lack explanation in places and appropriate comparisons to existing work. The paper is missing references to several key pieces of related work tackling similar problems. I identified (fixable) issues with the theoretical results in the paper and felt that overall the paper was rushed and difficult to read in places as a result.

Further, this paper exceeds the recommended 8 page limit with ill-formatted references.


Overall

1) Claim that threat models beyond l2 has not been explored before is false. See Anil et al. for networks implementing an L-infinity Lipschitz constraint [1] and Huster et al. [2] for a discussion of infinity norm constrained architectures (the latter is cited).

2) In this work you claim through empirical evidence that adversarial training has only a small effect on the upper bound of the Lipschitz constant. This is contested by e.g. [3] which focused on measuring the Lipschitz constant of neural networks tightly and highlighted the impact of adversarial training.

3) Ignoring issues discussed below, this paper presents an interesting theoretical result on providing stochastic robustness guarantees for Lipschitz constrained neural networks.  However, the paper does not discuss existing baselines which provide similar stochastic guarantees e.g. [4].


4) There are several (fixable) issues present in the theoretical results. First, notation is inconsistent throughout with the loss function $l$ taking either one or two arguments in different places (e.g. Eqn. 2 vs. Proposition 1). In fact, in Proposition 1 the loss function on the left hand side of the inequality takes two arguments not one (one of which is vector valued).

From the proof, it is clear that the authors mean to make the loss a function of the margin (which should be discussed clearly). Further, Proposition 2 presents the wrong terms in the sum (so that E[L] != adversarial risk). L should instead consider the sum over the maximum change in the margin for the dataset and use Proposition 1 after Eqn 11 as described. Further, this proof is a straight forward application of McDiarmid's inequality and does not need to define a Doob martingale (in fact, it is not clear that one is defined here as no filtration is presented). I also believe that the presented bound is computed incorrectly, as the maximum change in the sum value through modifying a single element would be B/n . I believe the stated result can be made rigorous and correct, however in its current form there are mistakes.

5) I do not understand the proposed training strategy. It seems as though standard cross entropy training is used but with a modified Lipschitz constraint which allows the output of the vector-valued function to have different Lipschitz constants at each index. If this is the case, could you please elaborate on what you mean by the argument that subnetworks associated with each class can achieve higher accuracy?

6) In section 4.1 you introduce the methods use to constrain the Lipschitz constant of the network. The first of which is a regularization scheme which provides no provable guarantees on the Lipschitz constant itself. Further, you describe spectral normalization as a regularization technique which is inaccurate --- it is typically implemented as a projection. Spectral normalization is also unable to tightly enforce the required Lipschitz constant (see e.g. [1]). Finally, I would like to ask how the authors compute the Lipschitz constants of the networks used throughout the experiments. Is the product of the linear layer p-norms computed after training? If so, this would give a loose upper bound (see [3]).

7) I am also concerned by the range of perturbation size used to measure the theoretical robustness of the networks -- a maximum perturbation size on CIFAR-10 of 4/255 is used which is very small compared to e.g. [4] which searches up to a radius 1.5.


I would consider raising my score if the issues present in the theoretical results are addressed by the authors.

Minor:

- In introduction: "so-called adversarial examples, appear to humans as normal images" may be considered to strict a definition. For example, images which look like random noise can force high classification confidence in classifiers and are often described as adversarial examples. Similarly, the thread model defined in equation (2) is presented as the general adversarial objective but is only a special case. One should be also be concerned with threat models violating a p-norm constaint.

- Page 5, typo "This means the the"

References:

[1] Sorting out Lipschitz function approximation, Cem Anil, James Lucas, and Roger Grosse
[2] Limitations of the Lipschitz constant as a defense against adversarial examples, Todd Huster, Cho-Yu Jason Chiang, and Ritu Chadha
[3] Efficient and Accurate Estimation of Lipschitz Constants for Deep Neural Networks, Mahyar Fazlyab, Alexander Robey, Hamed Hassani, Manfred Morari, and George J. Pappas
[4] Certified Adversarial Robustness via Randomized Smoothing, Jeremy M Cohen, Elan Rosenfeld, and J. Zico Kolter

**Experience Assessment:**

I have published one or two papers in this area.

**Review Assessment: Checking Correctness Of Derivations And Theory:**

I carefully checked the derivations and theory.

**Review Assessment: Checking Correctness Of Experiments:**

I assessed the sensibility of the experiments.

**Review Assessment: Thoroughness In Paper Reading:**

I read the paper thoroughly.

---

### Official Review · AnonReviewer1 · 2019-10-23
**Official Blind Review #1**

**Rating:** 1

**Review:**

1. Contributions:
A) Extension of robustness bound based on margin and Lipschitz constant of the network to arbitrary l_p norms. Significance: Low.

B) High probability bound on adversarial robustness based on A) and McDiarmid's inequality. Significance: Low.

C) Lipschitz constant bound for one-versus-all networks (OVAs). Significance: Low.

D) Experimental evaluation. Significance: Low

2.  Detailed comments:
Originality: Contribution A) is a simple extension of previously known bounds for the l_2 norm which is rather technical. As such It has questionable novelty and limited potential impact. The same goes for B) which is a simple application of a well-known concentration bound that holds for many statistical estimators based on averaging.

Also C) is a well-known alternative for multiclass classification (one-vs-all approach) and somehow defeats the purpose of representation reuse from intermediate layers of the network to perform classification.  In OVAs the prediction for each class depends on entirely independent values of hidden layers. As such it is not surprising that. a slightly tighter estimate of robustness can be obtained, as one does not have to worry about interactions between features from each independent sub-network.

Quality: The paper is in general well written, but lacks any depth or clear important contribution. The propositions are rather technical and/or simple extensions. The proposal of OVA networks and the bound on the adversarial risk are two parallel ideas that could be explored and hopefully find a significant contribution.

For example, what could the authors say about local Lipschitz constant estimates? are they easy/difficult to compute? what new bounds could be derived in term of such constants? For OVA networks can there be some intermediate reuse of hidden layers and still get some improvement in certified robustness? perhaps only the final layers require a split to see the empirical improvement, in this way reducing the burden of training one network from scratch for each different class (ImageNet has 1000 classes so OVAs could be quite expensive to train).

Or maybe for OVAs the complexity of each subnetwork can be substaintially less (e.g., less layers) to obtain the same accuracy/robustness?? It looks from the experiments that this is true but some theoretical insights about why it is the
case can yield a strong paper.

Clarity: The paper could see some improvements in notation and some erroneous claims:
1. Authors claim that the robustness can not be assessed when f is non-convex. This is in general, false. There is a constrained maximization procedure. constrained maximization of convex functions is a hard problem.  Not to be
confused with certain instances of constrained minimization of convex functions that can be solved efficiently. The same in section 5.1. The true statement is that for SVM the problem becomes optimization of a linear function with some simple constraint.

2. In section 3 it is not clear if the loss l depends on the label y or not. sometimes it appears as an argument, and sometimes it disappears. see for example equation (4).

3. In proposition 2 the assumption is that l is monotonically decreasing and its range is [0, B). Does this mean that B is negative?? this makes the claims in this section seem weird. There is some problem here. Also McDiarmids does not require the Doob Martingale concept, which is not even defined or referenced.

Significance: I find the significance low as the work is incremental and has technical rather than theoretical statements, and lacks a clear contribution. I vote for rejecting this submission.


**Experience Assessment:**

I have published one or two papers in this area.

**Review Assessment: Checking Correctness Of Derivations And Theory:**

I carefully checked the derivations and theory.

**Review Assessment: Checking Correctness Of Experiments:**

I assessed the sensibility of the experiments.

**Review Assessment: Thoroughness In Paper Reading:**

I read the paper thoroughly.

---

### Decision · Program_Chairs · 2019-12-19

**Decision:**

Reject

**Comment:**

The authors propose a novel method to estimate the Lipschitz constant of a neural network, and use this estimate to derive architectures that will have improved adversarial robustness. While the paper contains interesting ideas, the reviewers felt it was not ready for publication due to the following factors:

1) The novelty and significance of the bound derived by the authors is unclear. In particular, the bound used is coarse and likely to be loose, and hence is not likely to be useful in general.

2) The bound on adversarial risk seems of limited significance, since in practice, this can be estimated accurately based on the adversarial risk measured on the test set.

3) The paper is poorly organized with several typos and is hard to read in its present form.

The reviewers were in consensus and the authors did not respond during the rebuttal phase.

Therefore, I recommend rejection. However, all the reviewers found interesting ideas in the paper. Hence, I encourage the authors to consider the reviewers' feedback and submit a revised version to a future venue.